# Synthesis of Tris-pillar[5]arene and Its Association with Phenothiazine Dye: Colorimetric Recognition of Anions

**DOI:** 10.3390/molecules24091807

**Published:** 2019-05-10

**Authors:** Alena Khadieva, Vladimir Gorbachuk, Dmitriy Shurpik, Ivan Stoikov

**Affiliations:** Kazan Federal University, Kremlevskaya, 18 Kazan, Russia; as-alex93@mail.ru (A.K.); leongard87@mail.ru (V.G.); dnshurpik@mail.ru (D.S.)

**Keywords:** pillar[5]arene, complex, phenothiazine, anion, multipillar[5]arene

## Abstract

A multicyclophane with a core based on tris(2-aminoethyl)amine (TREN) linked by amide spacers to three fragments of pillar[5]arene was synthesized. The choice of the tris-amide core allowed the multicyclophane to bind to anion guests. The presence of three terminal pillar[5]arene units provides the possibility of effectively binding the colorimetric probe *N*-phenyl-3-(phenylimino)-3H-phenothiazin-7-amine (**PhTz**). It was established that the multicyclophane complexed **PhTz** in chloroform with a 1:1 stoichiometry (lgK_a_ = 5.2 ± 0.1), absorbing at 650 nm. The proposed structure of the complex was confirmed by ^1^H-NMR spectroscopy: the amide group linking the pillar[5]arene to the TREN core forms a hydrogen bond with the **PhTz** imino-group while the pillararenes surround **PhTz**. It was established that the **PhTz**:tris-pillar[5]arene complex could be used as a colorimetric probe for fluoride, acetate, and dihydrogen phosphate anions due to the anion binding with proton donating amide groups which displaced the **PhTz** probe. Dye displacement resulted in a color change from blue to pink, lowering the absorption band at 650 nm and increasing that at 533 nm.

## 1. Introduction

Creating supramolecular systems capable of anion detection is an important direction for modern chemistry. Contamination of the environment [1] with fertilizers, industrial byproducts (phosphates, nitrates, hydrosulfates) has led to the necessity of creating systems which are capable of recognizing anionic analytes; anions also play an important role in medicine [2]. Detecting inorganic anions allows the control of food quality [3]. In contrast to the sensors toward cations, the goal of creating sensors for inorganic anions remains unsolved. Low charge density and the large size of anions compared to cations are the main hindrances for creating selective sensors of anions. The introduction of amide, hydroxyl, urea, and thiourea fragments into receptor structures allow for the formation of additional coordination sites for more effective and selective binding of anionic substrates by hydrogen bonds [4,5,6]. One of the most widespread approaches to constructing macrocycle-derived anion sensors is using a macrocyclic platform for spatial preorganization of proton donating groups, which provides conformity of the receptors spatial structure to the substrate [7]. The anion is bound by forming hydrogen bonds with spatially preorganized functional groups [8,9] and with macrocyclic fragments [10,11].

Pillar[n]arenes are capable of anion recognition [12], forming supramolecular complexes [13], and they may serve as components of sensors [14,15]. They form inclusion complexes and associates with aromatic compounds [16], which makes them suitable structure blocks for receptor systems working on the dye-displacement principle for inorganic anions [17] and biologically relevant anion substrates, e.g., adenosine triphosphate [18]. However, the cavity of pillar[5]arene is too small for effective binding of most of the redox-active substrates. Only relatively small fragments are bound within the pillar[5]arene cavity with effectiveness. For example, alkyl groups and pyridinium fragments can enter into the pillar[5]arene cavity, while the pillar[n]arenes with the larger macrocycle size are still synthetically difficult to achieve [19]. This is the reason why we have proposed as an alternative approach to binding relatively large aromatic substrates to unify in one structure several pillar[5]arene fragments, which increase the effectivity of association with the aromatic substrate by multicenter interactions.

The synthesis of multicyclophanes is non-trivial from the point of view of organic chemistry. Several approaches to the synthesis of multicyclophanes based on thiacalixarene [20,21], pillararene [22], and of hybrid structures containing both fragments [23] in high yields have been developed previously in our group. It was shown that synthesized multicyclophanes are capable of forming stable supramolecular associates [20,21] and also can participate in oxidative polymerization reactions with supramolecular assistance [23]. TREN as a convenient platform for the design of anion receptors and sensors [24,25] has been chosen as a core of the synthesized multicyclophane. Pillararenes can effectively interact with phenazines, e.g., percarboxylated pillar[6]arene can form an inclusion complex with methylene blue in water with a high association constant [26]. Pillar[5]arenes as guest-molecules can form complexes with C-shaped strips formed by several phenazine structural units [27]. We have proposed a third approach, i.e., surrounding of phenazine with pillar[5]arene fragments. As a compound belonging to the phenazine class, a derivative of phenothiazine, PhTz (*N*-phenyl-3-(phenylimino)-3H-phenothiazin-7-amine) was chosen as a structural analog of methylene blue, containing two phenyl groups [28]. Due to the redox-activity of phenothiazine derivatives, investigation of supramolecular ensembles with this compound offers the opportunity for further application of obtained systems in electrochemical sensors [29,30]. Formation of stable complexes of multicyclophane with phenazine derivatives, considering the wide variety of structural derivatives of phenazines (in particular, phenothiazines) [31] opens wide opportunities for constructing colorimetric [32] and electrochemical [33] sensors.

In this study, we have developed an approach to synthesize a multicyclophane with a TREN core. Multicyclophane was synthesized by aminolysis of pillar[5]arene ethyl ester derivative with TREN. It was established that association of PhTz, a structural analog of methylene blue, with synthesized tris-pillar[5]arene occurs in chloroform. The formation of the complex is accompanied by the color change of PhTz from pink to blue. The interaction of the complex with inorganic anions was monitored by UV-vis spectrophotometry. Competition of tris-pillar[5]arene with anions and PhTz changed the solution color from blue (complex of tris-pillararene with PhTz) to pink in the presence of F^−^, AcO^−^, or H_2_PO_4_^−^ anions. This work is a proof-of-concept for creating supramolecular sensor systems based on multicyclophanes and phenothiazine-based dyes based on the dye-displacement principle for detecting anions by the colorimetric method.

## 2. Results and Discussion

### 2.1. Synthesis of Tris-pillar[5]arene

Aminolysis of the cyclophane mono-ester derivatives with polyamine compounds [34] is a well-known reaction that helps achieve target product in high yields. The necessity to adjust the reagents ratio and reaction conditions and many possible byproducts make it one of the least widespread approaches to multicyclophane synthesis. Pillar[5]arene **3** was chosen as a synthon. It was obtained by the literary procedure [22] in two stages, i.e., sequential mono-*O*-demethylation of pillar[5]arene **1** and further alkylation of phenol group of the compound **2** with ethyl bromoacetate (Scheme 1). Then, the macrocycle **3** was introduced into the reaction with tris(2-aminoethyl)amine. The conditions of reaction and reagents ratio were adjusted to obtain target tris-pillar[5]arene **4**. The optimal yield was reached with a 3.3-fold excess of the compound **3** against TREN in the toluene/methanol mixture under reflux for 72 h. As a result of aminolysis, tris-pillar[5]arene **4** was obtained with a yield of 62%. Based on a thin layer chromatography (TLC), byproducts of aminolysis (mono- and bis-derivatives) were found and then separated by column chromatography on silica gel (CH_2_Cl_2_:propanol-2 = 10:3).

In the MALDI mass-spectrum of compound **4** on a 2,5-dihydroxybenzoic acid matrix (Appendix A), molecular ion peaks corresponding to protonated and cationized with sodium molecular ions of tris-pillar[5]arene **4** were observed. The individual product was established by TLC, the structure of the compound was confirmed by FTIR, ^1^H, and ^13^C-NMR spectroscopy, MALDI mass-spectrometry (Appendix A) and its composition by elemental analysis.

### 2.2. Study on the Interaction of Tris-pillar[5]arene **4** with PhTz

The interaction of the synthesized tris-pillar[5]arene **4** with **PhTz** was studied (Figure 1). Chloroform was specified as a solvent due to the solubility of both components. Based on UV-spectrophotometric titration (Figure 2), a 1:1 complex was formed with the association constant lgK_a_ = 5.2 ± 0.1. Due to the presence of two absorption bands at 533 nm (free **PhTz**) and 650 nm (complex **PhTz:4**) the titration was accompanied with visually detectable color change (533 nm—pink, 650 nm—blue).

An assumption was made, that hydrogen bonding of PhTz to the amide groups of tris-pillar[5]arene **4** led to lower electron density in phenothiazine fragment significantly stabilized by interaction with the pillar[5]arene fragments. The cause of red shift is due to the two factors, i.e., hydrogen bond of tris-pillar[5]arene amide fragments with the imino group of **PhTz** and interaction of **PhTz** with the pillar[5]arene fragments (Figure 1). Examples of such effects of shifts of the absorption band of the phenothiazine derivatives in their complex are known in the literature. Thus, a significant red shift of the absorption band was observed in the interaction with *d*-metal cations also explained by lowering electron density in phenothiazine fragment [35] (152 nm red shift), [36] (90 nm red shift). There are examples of hydrogen bond and aromatic system interactions in dimers influencing the UV-spectra of structural analogs of methylene blue [37].

This assumption is supported by the ^1^H-NMR spectroscopy (Appendix A). Most significant changes are observed in proton signals of ethylidene fragment of tris-pillar[5]arene **4** core, TREN (appearance of a new wide signal), in proton signals of oxymethylene fragments (appearance of new signals with upfield shift) and phenothiazine aromatic protons (upfield shift). Participation of pillar[5]arene fragment in forming of the complex with **PhTz** is confirmed by widening of corresponding aromatic proton signals. For further support of assumption, additional experiments in DMSO-*d_6_* were carried out: in the presence of **PhTz,** the signal of NH-protons of multicyclophane is shifted from 8.14 ppm to 8.40 ppm (Appendix A).

The effects described above in ^1^H-NMR spectra of **PhTz**: the tris-pillar[5]arene **4** complex are in accordance with previously reported ones for inclusion complexes involving the pillar[6]arene cavity [26]. The association constant obtained (lgK_a_ = 5.2 ± 0.1) is lower than described in the literature (lgK_a_ = 7.06), for the inclusion complex of methylene blue implemented into percarboxylated pillar[6]arene cavity in water. However, this difference is not so dramatic taking into account the fact that host and guest mentioned in [26] have opposite (negative and positive, correspondingly) charges.

### 2.3. Investigation of Complex PhTz:4 Interaction with Anions

The interaction of the **PhTz:4** complex with a 10-fold excess of the tetrabutylammonium salts (NBu_4_X, where X = Br^−^, NO_3_^−^, F^−^, H_2_PO_4_^−^, AcO^−^, Cl^−^, and I^−^) was investigated. The most prominent effect was observed in the case of fluoride, dihydrogen phosphate, and acetate anions when the absorption band at 650 nm disappeared. In the case of chloride and bromide anions, the effect on the intensity of absorption bands was negligible while in the case of iodide and nitrate, the intensity of absorption band at 650 nm was slightly lower, than that for the **PhTz**:**4** complex (Figure 3). Fluoride, acetate, and dihydrophosphate anions have a similar basicity and surface charge density, which explains the high sensitivity of dye-displacement effect in the presence of corresponding tetrabutylammonium salts. Among other anions studied, the slight sensitivity of PhTz-tris-pillar[5]arene complex to nitrate and iodide compared to smaller chloride and bromide anions can be explained by steric effect of bulky pillararene substituents.

Competitive binding with dihydrogen phosphate anion was confirmed by ^31^P-NMR spectroscopy. The experiment was conducted with 5 × 10^−3^ M solution of **4**, **PhTz**, and tetrabutylammonium dihydrogen phosphate. The chemical shift of the signal corresponding to the dihydrogen phosphate anion shifted by δ = 0.67 ppm upfield in the presence of tris-pillar[5]arene **4** and occurred as a widened signal with the center shifted upfield by δ = 0.19 ppm in the presence of **4** and **PhTz** (Appendix A).

Changes in the spectrum of the **PhTz:4** complex (Figure 4) were observed with 3.3 × 10^−6^ M dihydrogen phosphate and acetate anions and 6 × 10^−6^ M fluoride anion. Therefore, the proposed system has characteristics comparable to modern colorimetric probes. It is the first example of the multicyclophane–phenazine derivative complex which is sensitive to small concentrations of anions. This offers principally new opportunities for the supramolecular design of materials for colorimetric sensors.

## 3. Materials and Methods

### 3.1. General Experimental Information

All reagents and solvents were used directly as purchased or purified according to the standard procedures. Analytical TLC was carried out using commercial silica gel plates and visualization was performed with the short wavelength UV light (254 nm). Column chromatography was performed with silica gel 60 H, slurry packed. The ^1^H and ^13^C-NMR spectra were recorded on a Bruker Avance 400 spectrometer (400.17 MHz for H-atoms) for 3–5% solutions in CDCl_3_ and DMSO-*d_6_*. The residual solvent peaks were used as an internal standard. Elemental analysis was performed on the Perkin-Elmer 2400 Series II instruments. IR spectra were recorded with Spectrum 400 IR spectrometer (Perkin Elmer (Waltham, MA, USA)). Absorbance frequencies are expressed in reciprocal centimeters (cm^−1^). MALDI spectra were recorded using an Ultraflex III mass spectrometer with 2,5-dihydroxybenzoic acid as a matrix. The peaks of molecular ions are represented by the most abundant mass. Melting points were determined using the Boetius Block apparatus.

The UV measurements were performed with a Shimadzu UV-3600 instrument. Quartz cuvettes with an optical path length of 10 mm were used. Binding constants were determined from the analysis of the binding isotherms obtained by UV spectroscopy and fitted to a 1:1 stoichiometry of binding. The Bindfit application [38,39] was used to calculate the association constant of **4** with **PhTz**. Absorbance values at 535 and 650 nm were used. Stoichiometry models 1:1, 1:2, and 2:1 were tried and the best fit was attained for 1:1. Three independent experiments were carried out.

Macrocycles **1**, **2,** and **3** were synthesized by previously reported methods [22]. 7-Phenylamino-3-phenylimino-3*H*-phenothiazine **PhTz** were synthesized by previously reported methods [27] (see Appendix A for details, spectra are presented in Appendix A).

### 3.2. Synthesis of Tris-pillar[5]arene ***4***

Tris{*N*-2-(4-carbonylmethoxy-8,14,18,23,26,28,31,32,35 nonamethoxypillar[5]arene)aminoethyl}amine: to 0.57 g (0.69 mmol) of compound 3, a mixture of toluene (14 mL) and methanol (6 mL) was added, followed by the addition of 0.031 g of tris(2-aminoethyl)amine (0.21 mmol). The mixture was refluxed under vigorous stirring during 72 h. The mixture was evaporated on a rotary evaporator, chloroform (25 mL) was added, and washed with 2M HCl solution (2 × 20 mL) and 5% ammonia solution (2 × 20 mL). Then the chloroform layer was dried with anhydrous sodium sulfate. The obtained reaction mixture was separated with column chromatography (CH_2_Cl_2_:propanol-2 = 10:3).

Product yield 0.32 g (62%), m. p.: 154°C; ^1^Н-NMR (DMSO-d_6_, 400 MHz) δ: 2.57 (m, 6H, -N-CH_2_-CH_2_-N), 3.25–3.26 (m, 6H, -N-CH_2_-CH_2_-N), 3.66–3.69 (m, 111H, -O-CH_3_, -CH_2_-), 4.66 (s, 6H, -O-CH_2_-C(O)-), 6.73–6.83 (m, 30H, Ar-H), 8.14 (m, 3H, -NH-); ^13^C-NMR (DMSO-d_6_, 100 MHz) δ: 28.42, 28.97, 29.31, 36.71, 51.35, 55.41, 65.11, 113.09, 113.28, 113.39, 113.72, 115.29, 127.16, 127.33, 127.46, 127.54, 127.64, 128.37, 148.35, 148.57, 149.93, 150.25, 150.69, 168.34; IR (cm^−1^) ν_max_ = 2988 (CPh–H), 2932(CH_2_, -CH_2_CН_2_-), 1677 (C=О), 1498 (CH_2_), 1206 (CPh–O–CH_2_); [Anal. Calcd. for C_144_H_162_N_4_O_33_: C, 69.83; H, 6.59; N, 2.26; found: C, 68.31; H, 5.77; N, 1.33]; MS (MALDI–TOF, *m*/*z*): found *m*/*z* = 2476.2 [M + H]^+^ and 2496.0 [M + Na]^+^, C_144_H_162_N_4_O_33_ for 2476.12.

## 4. Conclusions

A method of tris-pillar[5]arene synthesis was developed through the aminolysis of an ethyl carboxylate derivative of pillar[5]arene with TREN. The use of a tris-pillar[5]arene–phenothiazine derivative complex as an anion sensor has been demonstrated for the first time. The interaction of tris-pillar[5]arene with **PhTz** through hydrogen bond formation with the amide group and stabilization with the electron-rich pillar[5]arenes leads to a significant red shift from 535 to 650 nm. It was shown that due to competitive complex formation with anions, this system is capable of a colorimetric response due to phenothiazine dye displacement. Thus the findings of this work can be used in the development of new sensory systems, especially redox-sensitive and electrochemical sensors based on the proposed complexes of phenothiazine derivatives with multipillararene.

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
