# Peer review of "Synthesis of Tris-pillar[5]arene and Its Association with Phenothiazine Dye: Colorimetric Recognition of Anions"

_molecules, 2019, doi:10.3390/molecules24091807_

Reviewer 1 Report

In general, this manuscript would benefit from a thorough proofreading by a native English speaker. There are some structural issues with sentences that make it difficult to understand the overall meaning of the manuscript. Some examples are shown below, but the authors should bear in mind that this is not an exhaustive list:

(1) Association complexes of the pillararene with the guest are described as “associates;” while technically true, this is not the generally acceptable phrase used for such complexes.

(2) 1H NMR spectroscopy is described as “NMR 1H spectroscopy,” which again, is non-standard for English written text.

(3) There are commas in locations where no comma should likely be written, for example in the sentence in the introduction which states, “has let to the necessity of creating systems, which are capable of recognizing anionic analytes.”

(4) In the introduction, the authors write “the goal of creating sensors for inorganic anions is far from the solution.” I think they mean that this goal is unsolved, but again, it is hard to understand.

(5) The authors use the term “high effectivity.” This is generally referred to as “effectiveness” or “efficacy.”

More concerning are some of the technical errors and ambiguities that are seen in the paper. For example:

(1) The authors refer to “differences in anion geometry” as a hindrance for the creation of selective anion sensors. I am not sure what this means. Differences from other analytes? Differences among themselves? The authors should explain and also clarify how this contributes to difficulties in sensor development.

(2) The authors state that “Macrocyclic platforms, as a rule of thumb, are used for spatial preorganization of proton donating groups and allow providing conformity of the receptors spatial structure to the substrate.” This is both an overgeneralization and highly inaccurate under certain conditions. Macrocyclic platforms can be proton donating, proton accepting, or relatively proton neutral.

(3) The authors refer to pillararenes as being able to bind both aromatic guests and anions, which is true, but fail to indicate how each component associates with different parts of the supramolecular host. This needs to be clarified. I am also not sure how these dual abilities are used to benefit the system reported herein.

(4) The authors introduce the idea of “electrochemically active substrates” but fail to explain what that definition is or how it relates to anion sensing overall.

(5) In the “Results and Discussion” section, the authors describe aminolysis of the cyclophane as “accessible regioselective.” This descriptor is unclear and needs to be clarified.

(6) In the same section, the following sentence has nonsensical sentence structure and needs to be revised: “The necessity to adjust reagents ratio and reaction conditions, many possible by-products make it one of the least widespread approaches to multicyclophane synthesis.”

The authors refer to “by-products” of aminolysis. All instances of the word “by-products” should be replaced with “byproducts” as one, non-hyphenated word.

In general, I am concerned by the lack of control studies with a mono-pillararene and bis-pillararene analogue. Without these control studies, it is impossible to elucidate the specific role of the three-pillararene architecture on the resulting UV-visible properties. Similarly, other core linkers in addition to the aliphatic amino TREN component should be studied in order to understand the specific contribution of this structural component as well.

The authors indicate that hydrogen bonding interactions between the guest and the host are an “assumption.” This is an assumption that is possible to test experimentally, at least in theory, using 1H NMR spectroscopy and related spectroscopic experiments to elucidate protons involved in hydrogen bonding. Were such experiments attempted? If so, briefly include some of the results here. If not, the authors should consider attempting some of those experiments prior to resubmission of the paper in order to prove or disprove this assumption.

More concerningly, the authors refer to the log Ka of 5.2 as a “high association constant.” This is not actually a high association constant compared to other pillararene components, some of which have been reported to be as high as 1013 M-1. The lack of context provided for the measured log Ka raises some concerns about the authors’ understanding of the broader pillararene field. The authors do indicate that “Examples of such effects of shifts of the absorption band of the phenothiazine derivatives in their complex are known in the literature,” but no literature references in support of this assertion are provided.

Overall, while the science reported herein has some merit and is certainly interesting, significantly more work needs to be done both experimentally as well as in terms of manuscript writing and revision in order to bring this paper to a publication-ready state. Major revisions are recommended.

Author Response

Response to Reviewer 1 Comments

Point 1: Association complexes of the pillararene with the guest are described as “associates;” while technically true, this is not the generally acceptable phrase used for such complexes.

Response 1: Term “associates” was substituted with the term “complexes” in text.

Point 2: 1H NMR spectroscopy is described as “NMR 1H spectroscopy,” which again, is non-standard for English written text.

Response 2: The term “NMR 1H spectroscopy” was substituted with the term “1H NMR spectroscopy” in text.

Point 3: There are commas in locations where no comma should likely be written, for example in the sentence in the introduction which states, “has let to the necessity of creating systems, which are capable of recognizing anionic analytes.”

Response 3: Excessive commas have been deleted from text.

Point 4: In the introduction, the authors write “the goal of creating sensors for inorganic anions is far from the solution.” I think they mean that this goal is unsolved, but again, it is hard to understand.

Response 4: Phrase was changed to “the goal of creating selective sensors for inorganic anions remains unsolved”.

Point 5: The authors use the term “high effectivity.” This is generally referred to as “effectiveness” or “efficacy.”

Response 5: The term “high effectivity” was substituted with “effectiveness”.

Point 6: The authors refer to “differences in anion geometry” as a hindrance for the creation of selective anion sensors. I am not sure what this means. Differences from other analytes? Differences among themselves? The authors should explain and also clarify how this contributes to difficulties in sensor development.

Response 6: We intended to point at the differences in geometry among inorganic anions, however, after considering this commentary, we decided to remove this statement from introduction. The differences in geometry of anions require different geometries of hosts, while low charge density complicates selective recognition. Therefore, the sentence “The differences in anion geometry, low charge density are the main hindrances for creating selective sensors to anions.” was substituted with “Low charge density, large size of anions, compared to cations, are the main hindrances for creating selective sensors to anions.”

Point 7: The authors state that “Macrocyclic platforms, as a rule of thumb, are used for spatial preorganization of proton donating groups and allow providing conformity of the receptors spatial structure to the substrate.” This is both an overgeneralization and highly inaccurate under certain conditions. Macrocyclic platforms can be proton donating, proton accepting, or relatively proton neutral.

Response 7: Indeed, we wanted to generalize only the most widespread approaches to implementing macrocycles only as hosts to inorganic anions. The sentence “Macrocyclic platforms, as a rule of thumb, are used for spatial preorganization of proton donating groups and allow providing conformity of the receptors spatial structure to the substrate.” was substituted with “One of the most widespread approaches to constructing macrocycle-derived anion sensors is using macrocyclic platform for spatial preorganization of proton donating groups, which allows providing conformity of the receptors spatial structure to the substrate”.

Point 8: The authors refer to pillararenes as being able to bind both aromatic guests and anions, which is true, but fail to indicate how each component associates with different parts of the supramolecular host. This needs to be clarified. I am also not sure how these dual abilities are used to benefit the system reported herein.

Response 8: While TREN-tris-acetylamide derivatives to the best of our knowledge have never been used to bind aromatic substrates (while being effective anion hosts), the pillar[n]arene fragments can effectively bind with aromatic guest.  Therefore we indicate the role of TREN – derived fragment as binding anions, while the pillararene fragments – as coordinating with phenothiazine derivatives.

Point 9: The authors introduce the idea of “electrochemically active substrates” but fail to explain what that definition is or how it relates to anion sensing overall.

Response 9: Electrochemically active substrates – are those capable of reversible oxidation – reduction processes, which allows (in future studies) - their incorporation in electrochemical sensors. We substituted the term with “redox-active substrates”.

Point 10: In the “Results and Discussion” section, the authors describe aminolysis of the cyclophane as “accessible regioselective.” This descriptor is unclear and needs to be clarified.

Response 10: Corresponding phrase was changed to “Aminolysis of the cyclophane mono-ester derivatives with polyamine compounds [34] is a well known reaction allowing to achieve target product in high yields”.

Point 11: In the same section, the following sentence has nonsensical sentence structure and needs to be revised: “The necessity to adjust reagents ratio and reaction conditions, many possible by-products make it one of the least widespread approaches to multicyclophane synthesis.”

Response 11: The sentence “The necessity to adjust reagents ratio and reaction conditions, many possible by-products make it one of the least widespread approaches to multicyclophane synthesis.” has been substituted with the following “The necessity to adjust reagents ratio and reaction conditions and many possible byproducts make it one of the least widespread approaches to multicyclophane synthesis”.

Point 12: The authors refer to “by-products” of aminolysis. All instances of the word “by-products” should be replaced with “byproducts” as one, non-hyphenated word.

Response 12: The term “by-products” was substituted with “byproducts”.

Point 13: In general, I am concerned by the lack of control studies with a mono-pillararene and bis-pillararene analogue. Without these control studies, it is impossible to elucidate the specific role of the three-pillararene architecture on the resulting UV-visible properties. Similarly, other core linkers in addition to the aliphatic amino TREN component should be studied in order to understand the specific contribution of this structural component as well.

Response 13: Preliminary experiments show no significant effect on UV-spectrum of phenothiazine derivative, upon addition of mono-pillararene analogue containing amide group.

Point 14: The authors indicate that hydrogen bonding interactions between the guest and the host are an “assumption.” This is an assumption that is possible to test experimentally, at least in theory, using 1H NMR spectroscopy and related spectroscopic experiments to elucidate protons involved in hydrogen bonding. Were such experiments attempted? If so, briefly include some of the results here. If not, the authors should consider attempting some of those experiments prior to resubmission of the paper in order to prove or disprove this assumption.

Response 14: Answer: In CDCl3 1H NMR spectra of tris-pillararene, the signal corresponding to amide group overlaps with other proton signals, and therefore for the 1H NMR spectra in CDCl3 we have based our assumption only on proton shifts of the signals of the ethylidene fragment. In DMSO d6 signal of NH proton of multicyclophane is observed at 8.14 ppm, which is shifted to 8.40 ppm in presence of phenothiazine derivative. Such slight change is evidently related to proton-acceptor properties of solvent. Also, it should be noted that broadening of all signals in 1H NMR spectrum additionally confirms complex formation. Therefore, we have added this information (on chemical shifts of amide group signals) into the manuscript and, corresponding spectra were added to supporting information. Sentence “For further support of assumption additional experiment in DMSO-d6 was carried out: in presence of PhTz the signal of NH-protons of multicyclophane is shifted from 8.14 ppm to 8.40 ppm (Figure S7)” was added to manuscript.

Point 15: More concerningly, the authors refer to the log Ka of 5.2 as a “high association constant.” This is not actually a high association constant compared to other pillararene components, some of which have been reported to be as high as 1013 M-1. The lack of context provided for the measured log Ka raises some concerns about the authors’ understanding of the broader pillararene field. The authors do indicate that “Examples of such effects of shifts of the absorption band of the phenothiazine derivatives in their complex are known in the literature,” but no literature references in support of this assertion are provided.

Response 15: Sentence “high association constant” are removed to the text. Examples were given only to the effects of hydrogen bonding with phenothiazine.

Reviewer 2 Report

Khadieva and colleagues report a sensor based on linked pillar[5]arene macrocycles and a hydrogen bonded dye which is displaced in the presence of specific aniond ro give a colorimetric response. The synthetic work is sound and all analysis appropriate for the systems studied. The conclusions are reasonable based on the data presented. Scientifically I have no problem with this, however, the English requires a little attention. Below I have noted most of the examples where this could be improved. Once these minor corrections have been made, this will be suitable for publication in Molecules.

Abstract

A multicyclophane with a core based on tris(2-aminoethyl)amine (TREN) linked by amide spacers to three fragments of pillar[5]arene was synthesized. The choice of the tris-amide core allowed the multicyclophane to bind to anion guests. The presence of three terminal pillar[5]arene units provides the possibility of effectively binding the colorimetric probe N-phenyl-3-(phenylimino)-3H-phenothiazin-7-amine (PhTz). It was established that the multicyclophane complexed PhTz in chloroform with a 1:1 stoichiometry (lgKa = 5.2±0.1) , absorbing at 650 nm. The proposed structure of the complex was confirmed by 1H NMR spectroscopy: the amide group linking the pillar[5]arene to the TREN core forms a hydrogen bond with the PhTz imino-group while the pillararenes surround PhTz. It was established that the PhTz:tris-pillar[5]arene complex could be used as a colorimetric probe for fluoride, acetate and dihydrogen phosphate anions due to the anion binding with proton donating amide groups which displaced the PhTz probe. Dye displacement resulted in a color change from blue to pink, lowering the absorption band at 650 nm and increasing that at 533 nm.

Introduction

“important direction of modern chemistry development. Contamination of environment” would be better as: “important direction for modern chemistry. Contamination of the environment” 

“allows controlling the food quality” would be better as: “allows the control of food quality” 

“is far from the solution.” would be better as: “is far from simple.” 

“allows forming additional coordinating sites” would be better as: “allows for the formation of additional coordination sites”

“The anion is bond by” should be: “The anion is bound by”

“macrocyclic fragment” should be: “macrocyclic fragments

“associates” should be: “complexes

“bound with pillar[5]arene cavity” should be:  “bound within the pillar[5]arene cavity”

“capable to form” should be: “capable of forming

“derivative of phenothiazine,” should be: “a derivative of phenothiazine,”

“synthesize multicyclophane on the base of TREN as a core” would be better as: “synthesize a multicyclophane with a TREN core

Results 

“is accessible regioselective, yet one of the non-trivial from a synthetic point of view approaches to synthesis of multicyclophanes.” would be better as: “is regioselectively accessible, yet non-trivial from a synthetic point of view.”

“conditions, many possible by-products” would be better as: “conditions, and many possible by-products,

“In MALDI mass-spectrum of the compound 4 on the 2,5-dihydroxybenzoic acid matrix” would be better as: “In the MALDI mass-spectrum of compound 4 on a 2,5-dihydroxybenzoic acid matrix”

“that hydrogen bond of PhTz with amide groups of” should be: “that hydrogen bonding of PhTz to the amide groups of” 

“The above-described effects” would be better as: “The effects described above

“Chemical shift of the signal corresponded to dihydrogen phosphate anion was shifted by δ=0.67 ppm to the upfield in the presence of” should be: “The chemical shift of the signal corresponding to the dihydrogen phosphate anion shifted by δ=0.67 ppm upfield in the presence of”

“with the center shifted by δ=0.19 ppm to the upfield” should be: “with the center shifted upfield by δ=0.19 ppm”

“of PhTz:4 complex (Figure 4) were observed at” should be: “of the PhTz:4 complex (Figure 4) were observed with

Conclusions

A method of tris-pillar[5]arene synthesis was developed through the aminolysis of an ethyl carboxylate derivative of pillar[5]arene with TREN. The use of a tris-pillar[5]arene - phenothiazine derivative complex as an anion sensor has been demonstrated for the first time. The interaction of tris-pillar[5]arene with PhTz through hydrogen bond formation with the amide group and stabilization with the electron-rich pillar[5]arenes leads to a significant red shift from 535 to 650 nm. It was shown that due to competitive complex formation with anions, this system is capable of a colorimetric response due to phenothiazine dye displacement. 

Author Response

Response to Reviewer 2 Comments

Point 1: Khadieva and colleagues report a sensor based on linked pillar[5]arene macrocycles and a hydrogen bonded dye which is displaced in the presence of specific anions to give a colorimetric response. The synthetic work is sound and all analysis appropriate for the systems studied. The conclusions are reasonable based on the data presented. Scientifically I have no problem with this, however, the English requires a little attention. Below I have noted most of the examples where this could be improved. Once these minor corrections have been made, this will be suitable for publication in Molecules.

Abstract

A multicyclophane with a core based on tris(2-aminoethyl)amine (TREN) linked by amide spacers to three fragments of pillar[5]arene was synthesized. The choice of the tris-amide core allowed the multicyclophane to bind to anion guests. The presence of three terminal pillar[5]arene units provides the possibility of effectively binding the colorimetric probe N-phenyl-3-(phenylimino)-3H-phenothiazin-7-amine (PhTz). It was established that the multicyclophane complexed PhTz in chloroform with a 1:1 stoichiometry (lgKa = 5.2±0.1), absorbing at 650 nm. The proposed structure of the complex was confirmed by 1H NMR spectroscopy: the amide group linking the pillar[5]arene to the TREN core forms a hydrogen bond with the PhTz imino-group while the pillararenes surround  PhTz. It was established that the PhTz:tris-pillar[5]arene complex could be used as a colorimetric probe for fluoride, acetate and dihydrogen phosphate anions due to the anion binding with proton donating amide groups which displaced the PhTz probe. Dye displacement resulted in a color change from blue to pink, lowering the absorption band at 650 nm and increasing that at 533 nm. 

Introduction

“important direction of modern chemistry development. Contamination of environment” would be better as: “important direction for modern chemistry. Contamination of the environment” 

“allows controlling the food quality” would be better as: “allows the control of food quality” 

“is far from the solution.” would be better as: “is far from simple.” 

“allows forming additional coordinating sites” would be better as: “allows for the formation of additional coordination sites”

“The anion is bond by” should be: “The anion is bound by”

“macrocyclic fragment” should be: “macrocyclic fragments

“associates” should be: “complexes

“bound with pillar[5]arene cavity” should be:  “bound within the pillar[5]arene cavity”

“capable to form” should be: “capable of forming

“derivative of phenothiazine,” should be: “a derivative of phenothiazine,”

“synthesize multicyclophane on the base of TREN as a core” would be better as: “synthesize a multicyclophane with a TREN core

Results 

“is accessible regioselective, yet one of the non-trivial from a synthetic point of view approaches to synthesis of multicyclophanes.” would be better as: “is regioselectively accessible, yet non-trivial from a synthetic point of view.”

“conditions, many possible by-products” would be better as: “conditions, and many possible by-products,

“In MALDI mass-spectrum of the compound 4 on the 2,5-dihydroxybenzoic acid matrix” would be better as: “In the MALDI mass-spectrum of compound 4 on a 2,5-dihydroxybenzoic acid matrix”

“that hydrogen bond of PhTz with amide groups of” should be: “that hydrogen bonding of PhTz to the amide groups of” 

“The above-described effects” would be better as: “The effects described above

“Chemical shift of the signal corresponded to dihydrogen phosphate anion was shifted by δ=0.67 ppm to the upfield in the presence of” should be: “The chemical shift of the signal corresponding to the dihydrogen phosphate anion shifted by δ=0.67 ppm upfield in the presence of”

“with the center shifted by δ=0.19 ppm to the upfield” should be: “with the center shifted upfield by δ=0.19 ppm”

“of PhTz:4 complex (Figure 4) were observed at” should be: “of the PhTz:4 complex (Figure 4) were observed with

Conclusions

A method of tris-pillar[5]arene synthesis was developed through the aminolysis of an ethyl carboxylate derivative of pillar[5]arene with TREN. The use of a tris-pillar[5]arene - phenothiazine derivative complex as an anion sensor has been demonstrated for the first time. The interaction of tris-pillar[5]arene with PhTz through hydrogen bond formation with the amide group and stabilization with the electron-rich pillar[5]arenes leads to a significant red shift from 535 to 650 nm. It was shown that due to competitive complex formation with anions, this system is capable of a colorimetric response due to phenothiazine dye displacement. 

Response 1: All amendments were added to the text.

Reviewer 3 Report

In this manuscript, the authors have designed a structure of associate with three units of pillar[5]arene as terminal fragments (N-phenyl-3-(phenylimino)-3H-phenothiazin-7-amine (PhTz)) for acting as bind colorimetric probe. This research topic is interesting and promising. However, there are some comments on the manuscript for further improvements.

In Page 5, Line 141, the authors mentioned that “Interaction of the PhTz:4 complex with a 10-fold excess of the tetrabutylammonium salts (NBu4X, where X=Br-, NO3-, F-, H2PO4-, AcO-, Cl-, I-) was investigated. The most prominent effect was observed in the case of fluoride, dihydrogen phosphate and acetate anions when absorption band at 650 nm totally disappeared. In the case of chloride and bromide anions, the effect on the intensity of absorption bands was negligible while in the case of iodide and nitrate the intensity of absorption band at 650 nm was slightly lower, than that for PhTz:4 complex (Figure 3). Here, they only described the result presented in “Figure 3. Absorption spectra of PhTz:4 complex (C = 1×10-5 M, PhTz:4, ratio 1:1) in the presence of the tetrabutylammonium salts (C = 1×10-4 M) and photographs of corresponding solutions”, and there is no explanation on the different absorption band induced by different salts.

Also, they indicated in Line 115 that “An assumption was made, that hydrogen bond of PhTz with amide groups of tris-pillar[5]arene 4 led to lower electron density in phenothiazine fragment significantly stabilized  by interaction with the pillar[5]arene fragments.” However, more supported data should be provided in the context of the manuscript.

Overall, more data and detailed discussion should be presented to support the conclusion of this research work.

Author Response

Response to Reviewer 3 Comments

Point 1: In Page 5, Line 141, the authors mentioned that “Interaction of the PhTz:4 complex with a 10-fold excess of the tetrabutylammonium salts (NBu4X, where X=Br-, NO3-, F-, H2PO4-, AcO-, Cl-, I-) was investigated. The most prominent effect was observed in the case of fluoride, dihydrogen phosphate and acetate anions when absorption band at 650 nm totally disappeared. In the case of chloride and bromide anions, the effect on the intensity of absorption bands was negligible while in the case of iodide and nitrate the intensity of absorption band at 650 nm was slightly lower, than that for PhTz:4 complex (Figure 3).” Here, they only described the result presented in “Figure 3. Absorption spectra of PhTz:4 complex (C = 1×10-5 M, PhTz:4, ratio 1:1) in the presence of the tetrabutylammonium salts (C = 1×10-4 M) and photographs of corresponding solutions”, and there is no explanation on the different absorption band induced by different salts.

Response 1: Due to dye-displacement effect, the absorption band corresponding to PhTz:4 complex at 650 nm has slightly lower intensity in case of nitrate and iodide anions and completely disappears in case of fluoride, acetate, dihydrophosphate anions which is presented in figure 3. Absorption band at 535 nm corresponds to free PhTz. Differences in absorption band intensities are caused by different affinities of tris-pillar[5]arene to the anions studied. Corresponding text fragment was added to the manuscript “Fluoride, acetate and dihydrophosphate anions have similar basicity and surface charge density which explains high sensitivity of dye-displacement effect in presence of corresponding tetrabutylammonium salts. Among other anions studied, slight sensitivity of PhTz-tris-pillar[5]arene complex to nitrate and iodide compared to smaller chloride and bromide anions can be explained by steric effect of bulky pillararene substituents.”

Point 2: Also, they indicated in Line 115 that “An assumption was made, that hydrogen bond of PhTz with amide groups of tris-pillar[5]arene 4 led to lower electron density in phenothiazine fragment significantly stabilized  by interaction with the pillar[5]arene fragments.” However, more supported data should be provided in the context of the manuscript.

Response 2: In 1H NMR spectra of tris-pillararene, the signal corresponding to amide group overlaps with other proton signals in CDCl3. Therefore we have based our assumption only on proton shifts of the signals of the ethylidene fragment. Additional experiment was carried out. In DMSO-d6 signal of NH proton of multicyclophane is observed at 8.14 ppm, which is shifted to 8.40 ppm in presence of phenothiazine derivative. Such slight change is evidently related to proton-acceptor properties of solvent. Also, it should be noted that broadening of all signals in 1H NMR spectrum additionally confirms complex formation. Therefore, we have added this information (on chemical shifts of amide group signals) into the manuscript and, corresponding spectra were added to supporting information. Sentence “For further support of assumption additional experiment in DMSO-d6 was carried out: in presence of PhTz the signal of NH-protons of multicyclophane is shifted from 8.14 ppm to 8.40 ppm (Figure S7)” was added to manuscript.

Point 3: Overall, more data and detailed discussion should be presented to support the conclusion of this research work.

Response 3: 1H NMR spectra in DMSO-d6 proving participation of amide fragment in forming hydrogen bond with PhTz were added to Supplementary information (Figure S7). Detailed discussion was added to manuscript. 

Round  2

Reviewer 1 Report

The authors have done a reasonable job responding to my comments in the previous round of reviews. A thorough proofreading by a native English speaker is still recommended, but otherwise the manuscript is fine for publication.

Reviewer 3 Report

The authors have revised the context of the manuscript. However, there is no results to be added into the main manuscript to further support the research conclusion.